# GENERATIVE ADVERSARIAL NEURAL ARCHITECTURE SEARCH WITH IMPORTANCE SAMPLING

## ABSTRACT

Despite the empirical success of neural architecture search (NAS) in deep learning applications, the optimality, reproducibility and cost of NAS schemes remain hard to assess. The variation in search spaces adopted has further affected a fair comparison between search strategies. In this paper, we focus on search strategies in NAS and propose Generative Adversarial NAS (GA-NAS), promoting stable and reproducible neural architecture search. GA-NAS is theoretically inspired by importance sampling for rare event simulation, and iteratively refits a generator to previously discovered top architectures, thus increasingly focusing on important parts of the search space. We propose an efficient adversarial learning approach in GA-NAS, where the generator is not trained based on a large number of observations on architecture performance, but based on the relative prediction made by a discriminator, thus significantly reducing the number of evaluations required. Extensive experiments show that GA-NAS beats the best published results under several cases on the public NAS benchmarks including NAS-Bench-101, NAS-Bench-201, and NAS-Bench-301. We further show that GA-NAS can handle ad-hoc search constraints and search spaces. GA-NAS can find new architectures that enhance EfficientNet and ProxylessNAS in terms of ImageNet Top-1 accuracy and/or the number of parameters by searching in their original search spaces.

## 1 INTRODUCTION

Neural architecture search (NAS) improves neural network model design by replacing the manual trial-and-error process with an automatic search procedure, and has achieved state-of-the-art performance on many computer vision tasks (Elsken et al., 2018). Since the underlying search space of architectures grows exponentially as a function of the architecture size, searching for an optimum neural architecture is like looking for a needle in a haystack. A variety of search strategies have been proposed for NAS. Typical search strategies include random search (Li & Talwalkar, 2020), differentiable architecture search and optimization (e.g., DARTS (Liu et al., 2018), SNAS (Xie et al., 2018), Bayesian optimization (e.g., NASBOT (Kandasamy et al., 2018)), and reinforcement learning (e.g., ENAS (Pham et al., 2018), NASNet (Zoph et al., 2018)).

In spite of a proliferation of NAS strategies proposed, issues on the robustness and reproducibility of existing NAS methods are raised by Li & Talwalkar (2020), Yu et al. (2019), and Yang et al. (2019). Comparisons between different methods are hard, as there is no shared search space or experimental protocol followed by all (Yu et al., 2019; Yang et al., 2019). To promote reproducibility and fair comparisons among methods, multiple NAS benchmarks have recently emerged, including NAS-Bench-101 (Ying et al., 2019), NAS-Bench-201 (Dong & Yang, 2020), and NAS-Bench-301 (Siems et al., 2020), which are datasets that map architectures to their evaluation metrics. This provides an opportunity for researchers to fairly benchmark different search algorithms in terms of searching for the highest-ranked architectures within the least number of queries to architecture performance, the latter being a good indicator of the search cost. However, Yu et al. (2019) has evaluated several well-known search algorithms, including DARTS (Liu et al., 2018), ENAS (Pham et al., 2018) and NAO (Luo et al., 2018) on NAS-Bench-101, and has concluded that these search algorithms perform similarly to a random policy (Yu et al., 2019).

In this paper, we revisit the NAS problem with *importance sampling*, a robust method for rare event discovery with theoretical guarantees (Rubinstein & Kroese, 2016; 2013). Our method stems from the

*Cross Entropy* (CE) method (Rubinstein & Kroese, 2013), which iteratively retrains an architecture generator to mimic the winning architectures generated in previous iterations so that the generator will increasingly focus on more important regions of the large search space. However, as the generator obtains performance measurements only after architectures, which are discretized graphs, are sampled and evaluated, the performance signal cannot be backpropagated to train the generator directly with SGD. Another challenge is the large search space an algorithm, e.g., reinforcement learning (RL), must explore before it can learn a policy to generate better performing architectures, which leads to high search cost.

To solve these challenges, we propose Generative Adversarial NAS (GA-NAS), an efficient and robust search algorithm for NAS. GA-NAS uses RL to train an architecture generator network based on RNN and GNN, avoiding the end-to-end differentiability issue. In contrast to other RL-based NAS schemes, GA-NAS does not obtain rewards from the performance metrics of the generated architectures, a costly procedure if a large number of architectures are explored. Instead, it iteratively updates a discriminator that can distinguish the currently top architectures from randomly generated ones, and uses the relative prediction from the discriminator to train the generator to sample even better architectures in the next round. This enables the generator to be efficiently trained without many queries to true architecture performance. We point out the theoretical connection between GA-NAS and a CE method with a symmetric Jensen–Shannon (JS) divergence loss, for which we establish the convergence guarantee.

Results from extensive search experiments show that GA-NAS outperforms state-of-the-art results reported the public benchmark sets on NAS-Bench-101, NAS-Bench-201, and NAS-Bench-301, in that it consistently finds top ranked architectures in fewer number of queries. We also demonstrate the ability of GA-NAS to incorporate hard ad-hoc constraints and can be used to improve existing models. Through experiments on ImageNet, we show that GA-NAS can enhance EfficientNet-B0 (Tan & Le, 2019) and ProxylessNAS (Cai et al., 2018) in their respective original search spaces, resulting in architectures with higher accuracy and/or smaller models.

## 2 RELATED WORK

A typical NAS method consists of the *search phase* and the *evaluation phase* (Yu et al., 2019). This paper is concerned with the search phase, of which the most important performance criteria are robustness, reproducibility (Li & Talwalkar, 2020) and search cost (Yu et al., 2019).

DARTS (Liu et al., 2018) has given rise to numerous optimization schemes for NAS (Xie et al., 2018; Chen et al., 2019; Xu et al., 2020; Chen & Hsieh, 2020; Li et al., 2020). While the objectives of these algorithms may vary, they all operate in the same or similar search space. However, Yu et al. (2019) demonstrates that DARTS performs similarly to a random search and its search results heavily dependent on the initial random seed. Furthermore, DARTS is criticized for converging to architectures with smooth loss landscapes which may not generalize well (Shu et al., 2019; Chen & Hsieh, 2020). In contrast, GA-NAS aims to train an architecture generator using the GAN framework while using importance sampling to gradually shift toward more important parts of the search space. GA-NAS has a convergence guarantee under certain assumptions. Its results are reproducible and are not sensitive to initial seeds.

NAS-Bench-301 (Siems et al., 2020) provides a formal benchmark for all $10^{18}$ architectures in the DARTS search space using surrogate trained on a subset of 50k architectures. Preceding NAS-Bench-301 are NAS-Bench-101 (Ying et al., 2019) and NAS-Bench-201 (Dong & Yang, 2020). Both of these benchmarks provide performance metrics in a tabular format and perform a fully exhaustive evaluation across all architectures in their search spaces, which contain $423k$ and $15k$ architectures, respectively. GA-NAS can find high-performing architectures in all three benchmarks and proves to be a highly robust algorithm that is not sensitive to search spaces.

Besides the cell-based search spaces (Pham et al., 2018; Liu et al., 2018), GA-NAS, as a search strategy, also applies to macro-search (Cai et al., 2018; Wu et al., 2019; Tan et al., 2019), which searches for an ordering of a predefined set of blocks. We show that GA-NAS can be used to improve EfficientNet (Tan & Le, 2019) by iteratively generating better ordering of the same set of MBConv blocks adopted in original EfficientNet.

On the other hand, a number of RL-based NAS methods have been proposed. ENAS (Pham et al., 2018) is the first Reinforcement Learning scheme in weight-sharing NAS. TuNAS (Bender et al., 2020) shows that guided policies decisively exceed the performance of random search on vast search spaces. AlphaX (Wang et al., 2020) uses Monte Carlo Tree Search to balance exploration with exploitation during search. Unlike these approaches, GA-NAS proves to be a highly efficient RL solution to NAS, since the rewards used to train the generator network comes from the relative performance prediction of the discriminator instead of from architecture evaluation. Our ablation studies show that the use of a discriminator can lower the number of architecture evaluations tremendously, which is typically a bottleneck in any NAS method.

Hardware-friendly NAS algorithms may take constraints such as model size, FLOPS, and inference time into account (Cai et al., 2018; Tan et al., 2019; Wu et al., 2019; Chen et al., 2020; Yu et al., 2020), usually by introducing regularizers into the loss functions (Wu et al., 2019; Cai et al., 2018) Contrary to these methods, GA-NAS can support ad-hoc search tasks, by enforcing customized hard constraints in importance sampling instead of resorting to approximate penalty terms.

## 3 THE PROPOSED METHOD

We describe importance sampling for rare event simulation, followed by a description of a particular implementation of it on NAS, which is the proposed GA-NAS method.

### 3.1 IMPORTANCE SAMPLING

We can view NAS as a combinatorial optimization problem. For example, suppose that $x$ is a Directed Acyclic Graph (DAG) connecting a certain number of operations, each chosen from a predefined operation set. Let $S(x)$ be a real-valued function representing the performance, e.g., accuracy, of $x$. In NAS, we aim to optimize $S(x)$ subject to $x \in \mathcal{X}$, where $\mathcal{X}$ denotes the underlying large search space of neural architectures.

One approach to solving a combinatorial optimization problem, especially the NP-hard ones, is to view the problem in the framework of *importance sampling* and *rare event simulation* (De Boer et al., 2005; Rubinstein & Kroese, 2013). In this approach we consider a family of probability densities $\{p(.;\theta)\}_{\theta \in \Theta}$ on the set $\mathcal{X}$, with the goal of finding a density $p(.;\theta^*)$ that assigns higher probabilities to the optimal solutions to the problem.

Assume $X$ is a random variable, taking values in $\mathcal{X}$ and has a prior probability density function (pdf) $p(.;\bar{\lambda})$ for fixed $\bar{\lambda} \in \Theta$. Let $S(X)$ be the objective function to be maximized, and $\alpha$ be a level parameter. Note that the event $\mathcal{E} := \{S(X) \geq \alpha\}$ is a *rare event* for an $\alpha$ that is equal to or close to the optimal value of $S(X)$. The goal of *rare-event* probability estimation is to estimate

$$l = l(\alpha) = \mathbb{P}_{\bar{\lambda}}(S(X) \geq \alpha) = \mathbb{E}_{\bar{\lambda}}\left[\mathbb{I}_{S(X) \geq \alpha}\right] = \int \mathbb{I}_{S(x) \geq \alpha} \, p(x;\bar{\lambda})dx,$$

where $\mathbb{I}_{x \in \mathcal{E}}$ is an indicator function that is equal to 1 if $x \in \mathcal{E}$ and 0 otherwise.

In fact, $l$ is called the *rare-event* probability (or expectation) which is very small, e.g., less than $10^{-4}$ (Botev et al., 2013). The general idea of *importance sampling* is to estimate the above rare-event probability $l$ by drawing samples $x$ from important regions of the search space with both a large density $p(x;\bar{\lambda})$ and a large $\mathbb{I}_{S(x) \geq \alpha}$, i.e., $\mathbb{I}_{S(x) \geq \alpha} = 1$. In other words, we aim to find $x$ such that $S(x) \geq \alpha$ with high probability.

Importance sampling estimates $l$ by sampling $x$ from a distribution $q^*(x, \alpha; \bar{\lambda})$ that should be proportional to $\mathbb{I}_{S(x) \geq \alpha} p(x; \bar{\lambda})$ (see (Murphy, 2012) page 820). Specifically, define the proposal sampling density $q(x)$ as a function such that $q(x) = 0$ implies $\mathbb{I}_{S(x) \geq \alpha} p(x; \bar{\lambda}) = 0$ for every $x$ (see (Botev et al., 2013) page 2). Then we have

$$l = \int \frac{\mathbb{I}_{S(x) \geq \alpha} p(x; \bar{\lambda})}{q(x)} q(x)dx = \mathbb{E}_q\left[\frac{\mathbb{I}_{S(X) \geq \alpha} p(X; \bar{\lambda})}{q(X)}\right]. \tag{1}$$

Rubinstein & Kroese (2016) shows that the optimal importance sampling probability density $q$ which minimizes the variance of the empirical estimator of $l$ is the density of $X$ conditional on the event

---

**Algorithm 1** JS-divergence minimization for rare-event estimation

---

1: $t \leftarrow 1, \rho_0 \leftarrow \rho, \theta_0 \leftarrow \bar{\lambda}$
2: **while** $\gamma(\theta_{t-1}, \rho_{t-1}) < \alpha$ **do**
3:      $\theta_t \in \arg\min_{\theta \in \Theta} JS(q(x, \gamma_{t-1}; \bar{\lambda}) || p(x; \theta))$, where $\gamma_{t-1} = \min(\alpha, \gamma(\theta_{t-1}, \rho_{t-1}))$ and

$$q(x, \gamma_{t-1}; \bar{\lambda}) = \frac{\mathbb{I}_{S(x) \geq \gamma_{t-1}} p(x; \bar{\lambda})}{\int \mathbb{I}_{S(x) \geq \gamma_{t-1}} p(x; \bar{\lambda}) dx}.$$

4:      Set $\rho_t$ such that $\gamma(\theta_t, \rho_t) \geq \min(\alpha, \gamma(\theta_{t-1}, \rho_{t-1}) + \delta)$ for some fix $\delta > 0$, and increment $t$.

---

$S(X) \geq \alpha$, that is

$$q^*(x, \alpha; \bar{\lambda}) = \frac{p(x; \bar{\lambda}) \mathbb{I}_{S(x) \geq \alpha}}{\int \mathbb{I}_{S(x) \geq \alpha} p(x; \bar{\lambda}) dx}. \tag{2}$$

However, noting that the denominator of (2) is the quantity $l$ that we aim to estimate in the first place, we cannot obtain an explicit form of $q^*(.)$ directly. To overcome this issue, Homem-de Mello & Rubinstein (2002) proposes the Cross-Entropy (CE) method to estimate $q^*(.)$ by generating a sequence of probability densities $p(.; \theta_1)$, $p(.; \theta_2)$, ..., that approaches $q^*(.)$ in terms of KL divergence (Homem-de Mello & Rubinstein, 2002), with details included in the Appendix.

Furthermore, motivated by the well-known fact that the symmetric Jensen-Shannon (JS) divergence is more robust than the asymmetric KL divergence (Nowozin et al., 2016), we replace the KL divergence with the JS divergence in our application of importance sampling (see the Appendix). Denote the threshold sequence by $\gamma_t, t \geq 0$, and sampling distribution parameters by $\theta_t, t \geq 0$. Initially, choose $\rho$ and $\gamma(\bar{\lambda}, \rho)$ so that $\gamma(\bar{\lambda}, \rho)$ is the $(1 - \rho)$-quantile of $S(X)$ under $p(.; \bar{\lambda})$, and generally, let $\gamma(\theta_t, \rho_t)$ be the $(1 - \rho_t)$-quantile of $S(X)$ under the the sampling density $p(x; \theta_t)$ of iteration $t$.

We have the following JS-divergence minimization rare-event estimation algorithm to generate a solution $x$ such that $S(x) \geq \alpha$ with high probability (with theoretical convergence guarantee presented in the Appendix):

In each iteration, we fit $\theta_t$ to minimize the JS divergence between the sampling density $p(x; \theta_t)$ and $q(x, \gamma_{t-1}; \bar{\lambda})$, the density of $X$ conditioned on that $S(X) \geq \gamma_{t-1}$. Since $\gamma_{t-1}$ is typically the $(1 - \rho_{t-1})$-quantile of $S(X)$ under the previous sampling density $p(x; \theta_{t-1})$, we are essentially fitting $\theta_t$ to the top candidates that had highest $S(x)$ in the previous iteration. Once the final sampling density is obtained, we can use that to generate a solution $x$ such that $S(x) \geq \alpha$ with high probability.

## 3.2 GENERATIVE ADVERSARIAL NEURAL ARCHITECTURE SEARCH

We now propose a generative adversarial training method named Generative Adversarial NAS (GA-NAS), as described in Algorithm 2, to replace the JS-divergence minimization (Step 3) in Algorithm 1, which is otherwise hard to implement. The GAN framework, as originally proposed by Goodfellow et al. (2014), alternates between training a discriminator $D$ between true and generated data and training a generator $G$ to minimize the probability that the discriminator distinguishes generated data from truth data. In other words, in adversarial learning, $D$ and $G$ play a two-player minimax game, which leads to the minimization of the JS-divergence between the distribution of truth data and that of the generator model (see Theorem 1 in (Goodfellow et al., 2014)). GA-NAS in Algorithm 2 alternates

---

**Algorithm 2** GA-NAS Algorithm

---

1: **Input:** An initial set of architectures $\mathcal{X}_0$; Discriminator $D$; Generator $G(x; \theta_0)$;
2: **for** $t = 1, 2, \dots, T$ **do**
3:      $\mathcal{T} \leftarrow$ top $K$ architectures of $\bigcup_{i=0}^{t-1} \mathcal{X}_i$ according to the performance evaluator $S(.)$.
4:      Let $G(x; \theta_{t-1})$ generate $k$ random architectures to form the set $\mathcal{F}$.
5:      Train discriminator $D$ with $\mathcal{T}$ being positive samples and $\mathcal{F}$ being negative samples.
6:      Train generator $G(x; \theta_t)$ using the output of $D(x)$ as the loss.
7:      Let $G(x; \theta_t)$ generate a new set of architectures $\mathcal{X}_t$.
8:      Evaluate the performance $S(x)$ of every $x \in \mathcal{X}_t$.

---

between training $D$ (Step 5) and $G$ (Step 6) to approximate the JS-divergence minimization between the distribution of top $K$ architectures $\mathcal{T}$ in iteration $t$ and the generator distribution $G(x; \theta_t)$. If $G(x; \theta_t)$ corresponds to $p(x; \theta_t)$, and the distribution of architectures in $\mathcal{T}$, i.e., the distribution of top architectures in prior iteration, corresponds to $q(x, \gamma_{t-1})$ in Algorithm 1, GA-NAS algorithm is essentially an implementation of the JS-divergence rare-event estimation. It replaces the JS minimization (Step 3) in Algorithm 1 by alternately learning $D$ and $G$, and instead of keeping track of $\rho_t$ and $\gamma_t$, using the top $K$ architectures as positive samples in each iteration. Roughly speaking, $\rho_t = \frac{k}{|\bigcup_{i=0}^{t-1} \mathcal{X}_i|}$, and $\gamma_t$ is chosen so that the $(1 - \rho_t)$-quantile is $\gamma_t$.

GA-NAS can operate on any search space. Here, we describe the implementation of the discriminator and generator in the context of cell search, whereas we evaluate GA-NAS on both cell search and macro search experiments. A *cell* architecture $\mathcal{C}$ is a Directed Acyclic Graph (DAG) consisting of multiple nodes and directed edges. Each intermediate node represents an operator, such as convolution or pooling, from a predefined set of operators. Each directed edge represents the information flow between nodes. We assume that a cell has at least one input node and only one output node.

**Pairwise Architecture Discriminator.** In the context of NAS, we often have a limited number of architectures with true accuracies. To efficiently use the truth data points, we construct the discriminator $D$ following a pairwise Siamese (Koch et al., 2015) scheme. For a pair of architectures, the discriminator $D$ determines the likelihood that the second architecture is from the same distribution as the first truth architecture. Thus, $D$ is a variant of the relativistic discriminator (Jolicoeur-Martineau, 2018). The discriminator is implemented by encoding both cells in the pair with a shared $k$-GNN model (Morris et al., 2019) followed by an MLP classifier with details provided in the Appendix.

**Architecture Generator.** An architecture is generated in an autoregressive fashion, which is a frequent technique in neural architecture generation such as in ENAS (Pham et al., 2018), NAO (Luo et al., 2018) and D-VAE (Zhang et al., 2019). At each time step $t$, given a partial cell architecture $\mathcal{C}_t$ generated by the previous time steps, GA-NAS uses an encoder-decoder architecture to decide what new operation to insert and which previous nodes it should be connected to. Similarly, the encoder is a multi-layer $k$-GNN. The decoder consists of a Feedforward-Softmax setup that outputs the operator probability distribution and a uni-directional Gated Recurrent Unit (GRU) (Chung et al., 2014) that recursively determines the edge connections to previous nodes.

**Training Procedure.** We train the GNN-based discriminator using pairs of architectures sampled from $\mathcal{T}$ and $\mathcal{F}$ based on supervised learning. We use Reinforcement Learning (RL) to train the architecture generator $G$ in a similar way to (You et al., 2018) for molecular generation. We associate the *action* at each time step (including the operation type and its connections to previous nodes) with an immediate reward $R_{step}$ based on its validity given the search space constraints. When the architecture generation terminates, a *final* reward $R_{final}$ penalizes the generated architecture $x$ according to the total number of violations of validity or rewards it with a score from the discriminator $D(x)$ that indicates how similar it is to the truth architectures. Both rewards together ensure that $G$ generates valid cells that are structurally similar to top cells from the previous time step. We adopt Proximal Policy Optimization (PPO) (Schulman et al., 2017), a policy gradient algorithm with generalized advantage estimation to train the policy.

The proposed learning procedure has several benefits. First, using the discriminator as a feedback mechanism can significantly reduce the number of architecture evaluations that are otherwise required in an RL-only scheme. Ablation studies demonstrate the effectiveness of the proposed discriminator in reducing the number of evaluations. Second, since the generator must sample a discrete architecture $x$ to obtain its loss on the discriminator $D(x)$, the entire generator-discriminator pipeline is not end-to-end differentiable and cannot be trained by SGD. Training $G$ with PPO solves this non-differentiability issue. Third, in PPO loss, there is an entropy loss term that encourages variations in the generated actions. By tuning the multiplier for the entropy loss, we can balance exploration/exploitation, which is crucial for a large search space. Please refer to the Appendix for a detailed discussion.

## 4   EXPERIMENTAL RESULTS

We verify the effectiveness of GA-NAS in two scenarios. First, we conduct experiments on several public NAS benchmarks under both non-weight-sharing and weight-sharing settings in order to fairly compare the powers of the search algorithms. Second, we demonstrate the use of GA-NAS to

improve a given neural architecture, including EfficientNet and ProxylessNAS, to achieve a higher accuracy and/or a lower number of parameters. We also refer interested readers to the Appendix for the ablation studies showing the effects of different components of GA-NAS on its performance.

## 4.1 PERFORMANCE ON NAS BENCHMARKS WITH OR WITHOUT WEIGHT SHARING

To evaluate search strategy alone and decouple it from variations of search spaces, we query three NAS benchmarks: NAS-Bench-101 (Ying et al., 2019), NAS-Bench-201 (Dong & Yang, 2020), and NAS-Bench-301 (Siems et al., 2020) and aim to find the best performing cell within the least number of queries. To further evaluate our algorithm when the true architecture accuracies are unknown, we train a weight-sharing supernet on NAS-Bench-101 and compare GA-NAS with a range of NAS schemes based on weight sharing.

**NAS-Bench-101** is the first publicly available benchmark for evaluating NAS algorithms. It consists of 423,624 DAG-style cell-based architectures, each trained and evaluated on CIFAR-10 (Krizhevsky, 2009) for 3 times. Metrics for each run include training time and accuracy. Querying NAS-Bench-101 corresponds to evaluating a cell in reality. We provide the results on two setups. In the first setup, we set $|\mathcal{X}_0| = 50$, $|\mathcal{X}_t| = |\mathcal{X}_{t-1}| + 50$, $t \geq 1$, and $K = 25$, In the second setup, we set $|\mathcal{X}_0| = 100$, $|\mathcal{X}_t| = |\mathcal{X}_{t-1}| + 100$, $t \geq 1$, and $K = 50$. For both setups, the initial set $\mathcal{X}_0$ is picked to be a random set, and the number of iterations $T$ is 10. We run each setup with 10 random seeds and average the results. The search cost is 8 GPU hours.

| Algorithm | Acc (%) | #Q |
|---|---|---|
| Random Search [†] | 93.66 | 2000 |
| RE [†] | 93.97 | 2000 |
| NAO [†] | 93.87 | 2000 |
| BANANAS | **94.23** | 800 |
| SemiNAS [†] | 94.09 | 2100 |
| SemiNAS [†] | 93.98 | 300 |
| **GA-NAS**-setup1 | **94.22** | **150** |
| **GA-NAS**-setup2 | **94.23** | **378** |

Table 1: The best accuracies found by different search algorithms on NAS-Bench-101 without weight sharing. Note that 94.23% and 94.22% are the accuracies of the 2nd and 3rd best cells. †: taken from (Luo et al., 2020)

| Algorithm | Mean Acc (%) | Mean Rank | Average #Q |
|---|---|---|---|
| Random Search | $93.84 \pm 0.13$ | 498.80 | 648 |
| Random Search | $93.92 \pm 0.11$ | 211.50 | 1562 |
| **GA-NAS**-Setup1 | $\mathbf{94.22 \pm 4.45e\text{-}5}$ | 2.90 | $647.50 \pm 433.43$ |
| **GA-NAS**-Setup2 | $\mathbf{94.23 \pm 7.43e\text{-}5}$ | 2.50 | $1561.80 \pm 802.13$ |

Table 2: The average statistics of the best cells found on NAS-Bench-101 without weight sharing, averaged over 10 runs (with std shown). Note that we set the number of queries (Q) for Random Search to be the same as the average number of queries incurred by GA-NAS.

Table 1 compares GA-NAS to other methods for the best cell that can be found by querying NAS-Bench-101, in terms of the accuracy and the rank of this cell in NAS-Bench-101, along with the number of queries required to find that cell. Table 2 shows the average performance of GA-NAS in the same experiment over multiple random seeds. Note that Table 1 does not list the average performance of other methods except Random Search, since all the other methods in Table 1 only reported their single-run performance on NAS-Bench-101 in their respective experiments.

In both tables, we find that GA-NAS can reach a higher accuracy in fewer number of queries, and beats the best published results, i.e., BANANAS (White et al., 2019) and SemiNAS (Luo et al., 2020) by an obvious margin. Note that 94.22 is the 3rd best cell while 94.23 is the 2nd best cell in NAS-Bench-101. From Table 2, we observe that GA-NAS achieves superior stability and reproducibility: GA-NAS-setup1 consistently finds the 3rd best in 9 runs and the 2nd best in 1 run out of 10 runs; GA-NAS-setup2 finds the 2nd best in 5 runs and the 3rd best in the other 5 runs.

| Algorithm | Mean Acc | Best Acc | Best Rank |
|---|---|---|---|
| DARTS [†] | $92.21 \pm 0.61$ | 93.02 | 57079 |
| NAO [†] | $92.59 \pm 0.59$ | 93.33 | 19552 |
| ENAS [†] | $91.83 \pm 0.42$ | 92.54 | 96939 |
| **GA-NAS** | $\mathbf{92.80 \pm 0.54}$ | **93.46** | **5386** |

Table 3: Searching on NAS-Bench-101 with weight-sharing, with the mean true test accuracy of the best cells from 10 runs, and the best accuracy/rank found by a single run. †: taken from (Yu et al., 2019)

To evaluate GA-NAS when true accuracy is not available, we train a weight-sharing supernet on the search space of NAS-Bench-101 (with details provided in Appendix) and report the true test

|  | CIFAR-10 | | | CIFAR-100 | | | ImageNet-16-120 | | |
|---|---|---|---|---|---|---|---|---|---|
| Algorithm | Mean Acc | Rank | #Q | Mean Acc | Rank | #Q | Mean Acc | Rank | #Q |
| REA [†] | 93.92 ± 0.30 | - | - | 71.84 ± 0.99 | - | - | 45.54 ± 1.03 | - | - |
| RS [†] | 93.70 ± 0.36 | - | - | 71.04 ± 1.07 | - | - | 44.57 ± 1.25 | - | - |
| REINFORCE [†] | 93.85 ± 0.37 | - | - | 71.71 ± 1.09 | - | - | 45.24 ± 1.18 | - | - |
| BOHB [†] | 93.61 ± 0.52 | - | - | 70.85 ± 1.28 | - | - | 44.42 ± 1.49 | - | - |
| RS-500 | 94.11 ± 0.16 | 30.81 | 500 | 72.54 ± 0.54 | 30.89 | 500 | 46.34 ± 0.41 | 34.18 | 500 |
| **GA-NAS** | **94.34 ± 0.05** | **4.05** | **444** | **73.28 ± 0.17** | **3.25** | **444** | **46.80 ± 0.29** | **7.40** | **445** |

† The results are taken directly from NAS-Bench-201 (Dong & Yang, 2020).

Table 4: Searching on NAS-Bench-201 without weight sharing, with the mean accuracy and rank of the best cell found reported. #Q represents the average number of queries per run. We conduct 20 runs for GA-NAS.

accuracies of architectures found by GA-NAS. We use the supernet to evaluate the accuracy of a cell on a validation set of 10k instances of CIFAR10 (see Appendix). Search time including supernet training is around 2 GPU days.

We report results of 10 runs in Table 3, in comparison to other weight-sharing NAS schemes reported in (Yu et al., 2019). We observe that using a supernet degrades the search performance in general as compared to true evaluation, because weight-sharing often cannot provide a completely reliable performance for the candidate architectures. Nevertheless, GA-NAS outperforms other approaches.

**NAS-Bench-201** contains 15,625 evaluated cells with DARTS-like (Liu et al., 2018) structures. The search space consists of 6 searchable edges and 5 candidate operations. We test GA-NAS on NAS-Bench-201 by conducting 20 runs for CIFAR-10, CIFAR-100, and ImageNet-16-120 using the true test accuracy. We compare against the baselines from the original NAS-Bench-201 paper (Dong & Yang, 2020) *that are also directly querying the benchmark data.* Since no information on the rank achieved and the number of queries is reported for these baselines, we also compare GA-NAS to Random Search (RS-500), which evaluates 500 unique cells in each run. Table 4 presents the results. We observe that GA-NAS outperforms all baselines on the task of finding the most accurate cell. Compared to RS-500, GA-NAS finds cells that are higher ranked while only exploring less than 3.2% of the entire search space in each run (querying only 445 out of 15,625 cells). It is also worth mentioning that in the 20 runs on all three datasets, GA-NAS can find the best cell in the entire search space more than once. Specifically for CIFAR-10, it found the best cell in 9 out of 20 runs [1].

**NAS-Bench-301** (Siems et al., 2020) is another recently proposed benchmark based on the same search space as DARTS. Relying on surrogate performance models, NAS-Bench-301 reports the accuracy of $10^{18}$ unique cells. We are especially interested in how the number of queries (#Q) needed to find an architecture with high accuracy scales in a large search space. We run GA-NAS on NAS-Bench-301 v0.9. We compare with Random (RS) and Evolutionary (EA) search baselines. Figure 1 plots the average best accuracy along with the accuracy standard deviations versus the number of queries incurred under the three methods. We observe that GA-NAS outperforms RS at all query budgets and outperforms EA when the number of queries exceeds 3k. The results on NAS-Bench-301 confirm that for GA-NAS, the number of queries

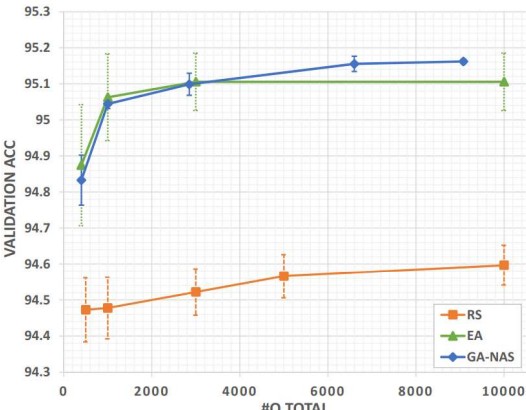

Figure 1: NAS-Bench-301 results comparing the means/standard deviations of the highest accuracy found at different total query limits.

(#Q) required to find a good performing cell scales well as the size of the search space increases. For example, on NAS-Bench-101, GA-NAS usually needs around 500 queries to find the 3rd best cell, with an accuracy ≈ 94% among 423k candidates, while on the huge search space of NAS-Bench-301 with up to $10^{18}$ candidates, it only needs around 6k queries to find an architecture with accuracy approximately equal to 95%.

---

[1] There are two cells in NAS-Bench-201 with the same, highest CIFAR-10 test accuracy of 94.37. We record that GA-NAS finds the best cell if either one is found.

| Algorithm | ResNet | | | Inception | | |
|---|---|---|---|---|---|---|
| | Best Acc | Train Seconds | #Weights (M) | Best Acc | Train Seconds | #Weights (M) |
| Hand-crafted | 93.18 | 2251.6 | 20.35 | 93.09 | 1156.0 | 2.69 |
| Random Search | 93.84 | 1836.0 | 10.62 | 93.14 | 1080.4 | 2.18 |
| **GA-NAS** | **93.96** | 1993.6 | 11.06 | **93.28** | 1085.0 | 2.69 |

Table 5: Constrained search results on NAS-Bench-101. GA-NAS can find cells that are superior to the ResNet and Inception cells in terms of test accuracy, training time, and the number of weights.

## 4.2 IMPROVING EXISTING NEURAL ARCHITECTURES

We now demonstrate that GA-NAS can improve existing neural architectures, including ResNet (He et al., 2016) and Inception (Szegedy et al., 2016) cells in NAS-Bench-101, EfficientNet-B0 (Tan & Le, 2019) under hard constraints, and ProxylessNAS-GPU (Cai et al., 2018) in unconstrained search.

For ResNet and Inception cells, we use GA-NAS to find better cells from NAS-Bench-101 under a lower or equal training time and number of weights. This can be achieved by enforcing a hard constraint in choosing the truth set $\mathcal{T}$ in each iteration. Table 5 shows that GA-NAS can find new, dominating cells for both cells, showing that it can enforce ad-hoc constraints in search, a property not enforceable by regularizers in prior work. We also test Random Search under a similar number of queries to the benchmark under the same constraints, which is unable to outperform GA-NAS.

We now use GA-NAS to improve the accuracy of well-known architectures found on ImageNet (Russakovsky et al., 2015), including EfficientNet-B0 and ProxylessNAS-GPU, which are already optimized strong baselines. For EfficientNet-B0, we set the constraint that the found networks all have an equal or lower number of trainable weights than EfficientNet-B0. For the ProxylessNAS-GPU model, we simply put it in the starting truth set and run an unconstrained search to further improve its top-1 validation accuracy. More details are provided in the Appendix. Table 6 presents the improvements made by GA-NAS over both existing models. Compared to EfficientNet-B0, GA-NAS can find new single-path networks that achieve comparable or better top-1 accuracy on ImageNet with an equal or lower number of trainable weights. We report the accuracy of EfficientNet-B0 and the GA-NAS variants *without data augmentation*. Total search time including supernet training is around 21 GPU days on Tesla V100 GPUs (20 GPU days for supernet training and 1 GPU day for the search).

For ProxylessNAS experiments, we train a supernet on ImageNet (Russakovsky et al., 2015) for around 20 GPU days, and conduct an unconstrained search using GA-NAS for around 38 hours on 8 Tesla V100 GPUs in the search space of ProxylessNAS (Cai et al., 2018), a major porition of which, i.e., 29 hours is spent on querying the supernet for architecture performance. Compared to ProxylessNAS-GPU, GA-NAS can find an architecture with a comparable number of parameters and a better top-1 accuracy on ImageNet.

| Network | #Params | Top-1 Acc |
|---|---|---|
| EfficientNet-B0 (no augment) | 5.3M | 76.7 |
| GA-NAS-ENet-1 | **4.6M** | 76.5 |
| GA-NAS-ENet-2 | **5.2M** | **76.8** |
| GA-NAS-ENet-3 | 5.3M | **76.9** |
| ProxylessNAS-GPU | 4.4M | 75.1 |
| GA-NAS-ProxylessNAS | 4.9M | **75.5** |

Table 6: Search results on the EfficientNet search space and ProxylessNAS search space.

## 5 CONCLUSION

In this paper, we propose Generative Adversarial NAS (GA-NAS), as a search strategy for NAS problems, based on a generative adversarial learning framework and importance sampling. Based on extensive search experiments performed on NAS-Bench-101, 201, and 301 benchmarks, we demonstrate the superiority of GA-NAS in finding more accurate architectures with much fewer queries which convert to evaluations in reality, as compared to a range of well-known NAS methods. We also show the capability of GA-NAS to improve existing architectures and its ability to search under ad-hoc hard constraints. GA-NAS improves EfficientNet-B0 by generating architectures in the same search space with higher accuracy and/or lower number of parameters and improves ProxylessNAS-GPU with enhanced accuracies and a slightly increased model size. These results indicate that GA-NAS generalizes well to diverse types of search spaces and can improve already optimized and strongly-performing neural architectures in their respective search spaces for large-scale image classification tasks.

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
