# OpenReview forum: "Generative Adversarial Neural Architecture Search with Importance Sampling"
_ICLR.cc/2021/Conference — Reject_

### Official Review · AnonReviewer1 · 2020-10-23

**Rating:** 4
**Confidence:** 4

**Review:**

In this paper, the authors introduce a NAS technique with an adversarial component. The discriminator learns to tell the difference between a set of good networks and randomly generated ones. This is quite a nice idea.

A few comments. I don’t think Section 3.1 adds anything and would be better off in an appendix. The connection between Algorithm 1 and Algorithms 2 seems fairly tenuous to me (although I could be wrong).

The bibliography is unacceptable. All papers with more than two authors are written as “et al.”, and there are glaring inconsistencies. arXiv is mentioned in multiple different ways and different fonts are used for different entries.
On a formatting issue, the text in the paper doesn’t look right compared to other ICLR submissions (it’s too pale). It would be worth looking in to this. The paper is otherwise fairly well written. Referring to EfficientNet as “Google’s EfficientNet” is quite odd. ResNet is not written as “Microsoft’s ResNet”.  I would recommend crediting the authors and not the institution. Table 2 appears above Table 1.

The details regarding training and the generator architecture are relegated to the appendix. These are very elaborate, which makes it very difficult to tell what is exactly contributing to the algorithm working. Table 7 seems to indicate that the discriminator itself can be removed for quite a small change in mean accuracy (94.2ish to 94.1ish). Without the discriminator the algorithm appears to be REINFORCE but with a more complicated generator network. On a related note, the comparison to REINFORCE is missing in table 1 which makes me suspect that this algorithm is basically the same thing in outcome, if not in effect.

Table 9 in the appendix (minor note — it’s not really an appendix when it’s in a separate file!) seems to show that different means of varying the size of the pool of networks over training has very little effect. (Having 94.227 in bold above 94.22 doesn’t change the fact that we are talking about 0.007%!)

The evaluation in Table 1 is very odd, as the authors are reporting the best acc instead of mean+-std as is common practice. This is unreliable, as when deploying these algorithms in the wild we are far more interested in how they do in expectation (particularly if they fail completely some of the time).

 Although the idea of using an adversarial framework to tell apart architectures in a search space is nice, the implementation has many moving parts, and doesn’t appear noticeably different to the standard REINFORCE NAS approach (which just has a generator as an RNN).  The presence of a discriminator has a very minimal effect, which is a shame. Some evaluation choices are very questionable. I am inclined towards rejection.

---

> ### Author Response · Authors · 2020-11-13
> **Connection between Algorithm 1 in Section 3.1 and Algorithm 2**
>
> Algorithm 1 in Section 3.1 presents the algorithmic steps of a general importance sampling framework for optimizing a black-box function, which is key to the proposed GA-NAS. In Section 3.2, we instantiate Algorithm 1 for our specific NAS problem with GAN and RL components. Step 3 of Algorithm 1 minimizes the JS-divergence between the two distributions in each iteration. It is well known that GAN solves a minimax problem in an iterative fashion and is equivalent to minimizing the JS-divergence between true data and generated data distributions. A particular implementation of the JS-minimizer (step 3) of Algorithm 1 is based on adversarial training and is described in Algorithm 2, where a discriminator and an RL-based generator are trained alternately in each iteration.  Furthermore, the relationship between the $\rho_t$ parameter in Algorithm 1 and the top-k parameter in Algorithm 2 is explained in Section 3.2.

---

> ### Author Response · Authors · 2020-11-13
> **Clarifying Tables, how to assess algorithms, and effectiveness of Discriminator**
>
> Thank you very much for your comments. We will fix the bibliography, formatting issues, the orders of tables and the naming of schemes. But there is some misunderstanding about how our algorithm works and is evaluated.
> 1. There is a misunderstanding about what Table 7 is presenting.  Ablation studies in Table 7 on Nas-Bench-101 show that without GAN (the discriminator), the algorithm degenerates into a GraphRNN RL policy trained by PPO (which is a more stable version of REINFORCE), as in RL-NAS-1, RL-NAS-2. Note that without GAN,  the RL only algorithm is much worse: 1) they cannot find top 3 architectures, they just found top 20, and top 900 architectures on average; 2) the result becomes less reproducible and less robust (see the std in acc, while our scheme has an almost zero std, proving guaranteed convergence); 3) most importantly, the # queries becomes very large, it takes 7000 queries to find top 20 architectures. The best accuracies do not differ much because we are searching on NAS-Bench-101 which only contain 423k architectures. Theoretically, any algorithm can find the best one in this dataset by exploring the space for enough large # of queries. That’s why for all these tables on NAS-Bench, please check the #Q (cost) together with the rank and the std of the accuracy achieved (robustness). Clearly, GAN is effective and is a key part of GA-NAS to reduce cost and promote robustness, according to Table 7.
>
> 2. Table 9 shows that without Linear Increase in $|\mathcal X_t|$, the best we can find is the 3rd best in NAS-Bench-101 no matter how many times we run it (the mean rank is 3 with a std of 0). With Linear Increase in $|\mathcal X_t|$ (Setup2), we can find 2nd best in 5 out of 10 runs and 3rd best in the other 5 runs. Moving from 94.22 to 94.227 is tiny, but moving from the 3rd best to 2nd best shows a big improvement of the capability of the algorithm---now it can find 2nd best which it could never find with a constant $|\mathcal X_t|$.
> By looking at the best rank a search algorithm can achieve in NAS-Bench-101, one can fairly verify the inherent power of the algorithm, removing the impact of other factors, e.g., the search space it operates in. For example, if you look Table 1 and Table 4, the other SOTA search algorithms are far behind GA-NAS in terms of the highest rank that can be found in a similar range of Queries. Furthermore, it is yet a big challenge to find The Best architecture within a similar range of Queries. Therefore, these ablation study Tables 7, 8, 9 clearly prove the usefulness of the components introduced.
>
> 3. Please note that in Table 2, we indeed report the mean and std information for 10 runs of our algorithm with 10 different random seeds. The numbers for GA-NAS in Table 1 and Table 2 come from the same set of experiments. We only list the best accuracies in Table 1 just to compare against other approaches, because for other approaches in the literature, only the best accuracy was reported; they didn’t report the mean/std performance. Thus, Table 1 is there just for a fair comparison. In fact, Table 1 is originally reported in [26], we have added the performance of GA-NAS and BANANAS to that table. We can see that the mean acc of GA-NAS is higher than the best accuracy of other schemes. The std is diminishing, demonstrating its superior robustness and reproducibility.
>
> The use of Adversarial Training and Discriminator helps to dramatically reduce the number of queries (which converts to evaluations in reality, no matter in weight-sharing or non-weight-sharing scenarios), therefore helping to dramatically reduce the search cost. This is shown in Table 7. It also increases the robustness and reproducibility as shown in Table 7. The proposed scheme is significantly different from REINFORCE by novel combination of GAN with importance sampling. Importance sampling is another mechanism that helps to reduce the number of queries made and increase robustness/reproducibility for the search algorithm as explained in Sec. 3.1. In fact, REINFORCE does not perform well according to Table 4. There is quite a big gap between REINFORCE and GA-NAS on this small 15k benchmark set of NAS-Bench-201.

---

### Official Review · AnonReviewer3 · 2020-10-28
**Not entirely convinced by the approach or premise**

**Rating:** 5
**Confidence:** 2

**Review:**

This paper proposes a Neural Architecture Search algorithm (GA-NAS) based on adversarial learning. The generator constructs architectures auto-regressively, which receives feedback from a GNN discriminator. Reinforcement learning (PPO) is used for training, to solve non-differentiability. GA-NAS’s effectiveness is demonstrated on several architecture search benchmarks for CIFAR-10 and 100, and is shown to improve EfficientNet for ImageNet.

Disclosure: I’m only vaguely familiar with the neural architecture search literature.

Pros:
1.	GA-NAS appears to consistently find high ranking architectures compared to the baselines, often at the cost of fewer queries.
2.	A fairly extensive set of experiments are included in the Experiments section and Appendix.
3.	GA-NAS is able to incorporate constraints into search. The authors demonstrate that GA-NAS is able to find a variation that slightly outperforms EfficientNet-B0.

Cons:
1.	As the name implies, the goal of NAS is search. GANs have proven excellent over the years at interpolation, but not extrapolation, which is what search/exploration requires. I’m concerned that a GAN-based approach is therefore limited in its search ability. How would GA-NAS compare with a random search policy constrained to be close to known “good” architectures?
2.	The GA-NAS generator and discriminator are initialized with an initial set of good architectures X_0. In the experiments, X_0 takes on the value of 50 and 100. The assumption that such a large number of “good” architectures are available ahead of time seems rather strong.
3.	The authors state that previous work has determined that other architecture search methods are not any better than random search. The gains in accuracies of the architectures produced by GA-NAS over the baseline seem moderate at best.

Questions:
1.	Where in the f-GAN paper is it stated that the JS divergence is more robust than the assymetric KL? Can you demonstrate this claimed advantage in GA-NAS by comparing objectives?
2.	Is X_0 counted in the number of queries reported? How were these initial architectures chosen? What rank are they?

Miscellaneous:
1.	This paper doesn’t adhere to the ICLR citation guidelines; citations in the text should include the first author’s last name and year. Additionally, the entire author list should appear in the references, not just “[First author] et al.” Please correct this during the rebuttal phase.
2.	Table 1: Typically, values in a table are bolded to represent the best values. BANANAS has the same accuracy value as GA-NAS.
3.	Why is the variance of GA-NAS’s accuracy so small in Table 1?
4.	Table 2 caption could be more descriptive. Besides the columns, how is this table differ from Table 1?
5.	There’s an extra space between “ImageNet” and footnote 2.

Rating:
While the results seem good, I’m not entirely convinced that a generative adversarial approach can effectively explore a neural architecture space, as the discriminator will inherently disincentivize deviating from previously seen architectures. I’m also concerned about the validity of the assumption that an initial set of known “good” architectures would be available to a search algorithm; the authors should clarify how these were selected. I lean toward rejection for now, but would be willing to raise my score if the authors could address my concerns.


========
Post-rebuttal
========

I thank the authors for answering my questions. While I am satisfied with the authors response to my concern about the "initial good architectures" assumption, I still remain unconvinced that adversarial learning can help search find new good architectures. I keep my score.

I also encourage the authors to carefully re-read the f-GAN paper, which explains exactly how any f-divergence (including the KL) can be implemented for adversarial learning. Switching to any f-divergence requires only a simple change to the loss function. It also appears that the authors significantly misunderstand VAEs. The difference between GANs and VAEs is not JS-divergence vs KL-divergence. Using a KL loss for adversarial learning does not require switching to a VAE. Given the central role they play in this paper's motivation, a better understanding of these subjects is important.

---

> ### Author Response · Authors · 2020-11-14
> **Explaining advantage of the symmetric JS Divergence**
>
> Thanks for mentioning the citation to f-gan paper. Regarding the JS-Divergence versus KL-Divergence, we need to cite two other papers. In fact, the differences between GANs (relying on JS-divergence) and Variational Auto Encoders (VAEs, which rely on KL divergence), in terms of comparing objectives, are comprehensively discussed on page 14 of “NIPS 2016 Tutorial: Generative Adversarial Networks” by Ian Goodfellow as well as page 2 of “TOWARDS PRINCIPLED METHODS FOR TRAINING GENERATIVE ADVERSARIAL NETWORKS” by Martin Arjovsky and L´eon Bottou. According to the discussion in the second article, asymmetric KL-divergence fails to work properly in two extreme cases. In the case $P_{real}(x) > P_{gen}(x)$, $P_{real}(x) > 0$, and $P_{gen}(x)$ goes to zero, the generator “does not cover parts of data”, and in the case $P_{gen}(x) > P_{real}(x)$ , $P_{gen}(x) > 0$, and $P_{real}(x)$  goes to zero, the generator generates “fake looking samples”. Therefore, it is a general belief in the literature that the symmetry of JS-divergence with respect to $P_{real}(x)$ and $P_{gen}(x)$ causes GANs to generate samples of better quality than VAEs (based on KL-divergence). This is one of the first few statements in GAN literature explaining the benefit of GAN, i.e., the adoption of JS-divergence, as compared to VAEs.
> However, we are afraid that we cannot easily switch to KL divergence objective for comparison. In fact, the GAN framework is minimizing the JS divergence. Switching to KL Divergence objective means that we need to replace the whole Generator-Discriminator GAN framework by VAE, which implements another completely different AutoEncoder framework.

---

> ### Author Response · Authors · 2020-11-14
> **Clarifying Initial Set, Exploration of GAN/PPO, and how to assess results**
>
> Thank you for pointing out the citation and formatting issues. We will fix those.
> 1.	Exploration ability of GA-NAS
>
> Your concern with the exploration capability of GAN is valid and reasonable. We would like to mention that although the original GAN has this limitation, when we combine it with a RL training setup, it gives us more control over the balance of exploration/exploitation. For instance, in the PPO training objective there is an entropy loss term in addition to the discriminator feedback, such that increasing this term would increase the variations in the selected action types leading to diversity in graph generation. With this loss term, we can encourage the generator to explore more by increasing its corresponding multiplier. In our experiments we tuned this multiplier and found that a value of 0.1 works the best. We’ll add more explanation regarding loss terms to make things clearer.
> We believe your suggested baseline, where we search with a random policy constrained to be close to known “good” architectures, is a simple idea. But it does not warrant enough exploration or guarantee convergence to optimality. The close-to-zero variance of GA-NAS in Table 2 indicates that GA-NAS can almost always converge to the corresponding accuracy (2nd or 3rd best) in the benchmark set, showing excellent stability, reproducibility and enough exploration ability.
>
> 2.	There is no assumption on the random initial set $\mathcal X_0$
>
> We would like to clarify that for our NAS-Bench-101, 201 and 301 experiments, we start with a totally random set of architectures $\mathcal X_0$ initially.  We do not assume starting with a set of “good” architectures at all. For example, if we set $|\mathcal X_0|$ to 50 for NAS-Bench-101 containing 423k unique architectures, it is unlikely that this initial random set contains any good ones. Then we select 25 top ones out of this initial 50 random ones to serve as the positive samples in discriminator training and kick start the algorithm. Therefore, GA-NAS does not rely on the starting set (which is totally random) and still converges consistently to top architectures, as demonstrated by low std of the accuracies achieved in multiple runs. In reality, GA-NAS is able to take full advantage of starting from some example networks. However, this is not a premise.
>
> Yes, the initial random set $\mathcal X_0$ is counted in the total number of queries incurred. If $|\mathcal X_0| = 50$, all initial 50 are evaluated and counted toward the total #queries. For NAS-Bench-101, the highest truth rank among the initial starting set is usually beyond 10k.
>
> 3.	Notes on assessing gains and benefits
>
> The gains of GA-NAS are not only to be assessed by the increase in mean accuracy, but should be read in the following ways: 1) the extremely low std means better reproducibility and stability; 2) It can always find the 2nd or 3rd best in NAS-Bench-101, regardless of the starting set (which are random architectures in the search space), whereas other algorithms didn’t even get into top-15. Similar observations can be made on NAS-Bench-201; 3) the number of queries incurred is lower, which indicates the search cost, no matter whether the evaluation is based on weight-sharing or train-from-scratch; 4) it can be used to improve existing network, e.g., EfficientNet, which is an already optimized strong baseline.
>
> All these imply that GA-NAS is designed to promote certainty, reproducibility and guaranteed convergence in search. It can move toward the direction desired instead of relying on heuristics and hoping for good results.  The certainty and robustness of the search results are achieved by importance sampling together with the exploration of RL plus GAN. Furthermore, the use of discriminator also reduces the search cost, since good/bad architectures are judged by the discriminator instead of by queries (evaluations).

---

### Official Review · AnonReviewer2 · 2020-10-28
**A flexible but complex and expensive NAS method.**

**Rating:** 5
**Confidence:** 4

**Review:**

A flexible but complex and expensive NAS method.

Summary:
The authors introduce a method for NAS that repeatedly trains a generator to sample candidate architectures. The method is evaluated on three NAS oracle benchmarks as well as constrained NAS settings. While there are some promising experimental results, I lean slightly against acceptance due to poor presentation, limited comparisons on most evaluations, and what seems like fairly limited benefits of the approach given its complexity and cost.

Strengths:
1. The method can easily incorporate constraints on computation and memory.
2. The method outperforms existing non-weight-sharing methods on several benchmarks.

Weaknesses:
1. The method introduces a lot of complexity such as a graph NN, a recurrent NN, and a full round of generative adversarial training in each search iteration. The computational cost of the latter is not discussed.
2. As with most non-weight-sharing methods, GA-NAS requires several hundred queries on each benchmark, which translates to GPU-weeks of search time. It is not clear that the benefits over weight-sharing methods, which are not quantified for most cases, outweigh this large search cost.
3. The results section for unconstrained search is confusing and it is hard to make comparisons (see notes 4-7 below).
4. From a look at the NAS-Bench-301 paper, it seems that BANANAS was the best non-weight-sharing method evaluated, but the authors compare only to EA and RS.
5. In the constrained search section, there are no comparisons to any other NAS methods. For example, random search is just as easy to apply to constrained problems as GA-NAS and should be used as a baseline.
6. There is no code in the supplementary materials. Will code be released?

Notes:
1. “Remain hard to be assessed” -> “Remain hard to assess”
2. The citation style does not follow ICLR guidelines.
3. “architectures are sampled, which are discretized graphs” -> “architectures, which are discretized graphs, are sampled”
4. What does it mean to “discover the Nth best architecture in Q queries”? Is “best” here according to test or validation? Why is Q a good metric for speed given that the algorithm doesn’t know to stop after query Q since in practice it won’t know the rank N of the current architecture?
5. Many numbers in paragraph 4 for Section 4.1 do not correspond to any number in any table.
6. Why isn’t the BANANAS performance bolded in Table 1?
7. Table 4 should include at least one weight-sharing method such as GDAS (Dong & Yang, 2019), which is much faster and performs reasonably well.

# Post-response update
Thank you to the authors for answering some of my questions and clarifying the search and evaluation of GA-NAS. I believe my original assessment that the contributed method was complex remains accurate; while the authors note that other methods like ENAS also use an RNN controller, in my view those methods are also complex. This paper increases this complexity with a GNN and an adversarial training setup. Use of such additions require showing significant improvements over baselines like random search, which I do not believe is achieved. I thus stand by my initial rating.

---

> ### Author Response · Authors · 2020-11-14
> **Complexity, Computational Cost, and Comparing with Weight Sharing Methods**
>
> 1. Regarding questions on complexity and computational cost, we would like to add that the search cost of GA-NAS is about 8 GPU hours on NAS-Bench-101 if we do not use weight-sharing, which is shorter compared to the time that would be spent on evaluations if architectures are not pre-labeled. With weight-sharing GA-NAS takes about 1 GPU day for search and 1 GPU day for supernet training on NAS-Bench-101 search space, making a total cost of 2 GPU days. We’ll add this information to the paper. In terms of network complexity, our GraphRNN generator has about the same complexity as some of the existing NAS works that use a popular RNN controller (ENAS, NAO). The only additional component here is an additional GNN encoder, which does not increase the training or inference time significantly and is able to better capture the useful features of a candidate architecture.
>
> 2. Regarding your concerns on comparing to BANANAS on NAS-Bench-301, when we conducted the NAS-Bench-301 experiment, we could only find a figure in the NAS-Bench-301 paper that reports the performance of BANANAS, since that figure has a y-axis with varying scales it is difficult to compare quantitatively. And we are not able to locate the official code that tests BANANAS on NAS-Bench-301, so it is not included as a baseline. However, we are able to locate the official BANANAS code for NAS-Bench-101 and hence we reported its performance there and believed that it would be enough.
>
> 3. Regarding your concerns on adding more baselines for constrained search, we agree with your point and are working on adding a RS baseline.
>
> 4. Regarding the code publication, it needs to be approved internally and we are looking into that. The code will be made public after all the paperwork is done.
> 5. Regarding your comments on adding GDAS in Table 4, we would like to clarify that in Table 4, we only test and compare with results obtained using non-weight-sharing as reported in NAS-Bench-201 paper, i.e., querying the benchmark for accuracy. We didn’t train a supernet for NAS-Bench-201 and didn’t include other methods reported in NB201 that relied on weight sharing. Since GDAS results in NAS-Bench-201 paper were obtained using weight-sharing methods, adding GDAS to Table 4 won't be a fair comparison to the listed methods.
> For weight sharing results, in addition to Table 3 on NAS-Bench101 and EfficientNet, we have an unconstrained-search experiment running GA-NAS on ProxylessNas search space with weight sharing, which achieves 75.52% final evaluation accuracy on full ImageNet, outperforming the original ProxylessNAS (75.1%). We did not add that result to the paper due to page limit, but we could add it to the manuscript, as more results on weight sharing.
> 6. Last but not least, we would like to thank you again for spotting the typos, formatting and style issues, these will be addressed.

---

> ### Author Response · Authors · 2020-11-14
> **Clarifying GA-NAS as a Search Strategy and #Queries as a Metric for Cost**
>
> Thank you for your constructive feedbacks and comments on our work. We would like to clarify our work and address some of your concerns.
> 1. Clarifying GA-NAS as a search strategy and weight-sharing as a performance estimator:
> We believe that there’s a slight misunderstanding regarding your comments on how GA-NAS compares with other weight-sharing methods. A typical NAS algorithm consists of a search algorithm and a performance estimation method. We would like to clarify that GA-NAS is a search algorithm and can work with any types of performance estimation methods, including weight-sharing supernet, a performance predictor (non-weight-sharing), or fully training each candidate architecture from scratch (non-weight-sharing). In fact, Table 3 presents the performance of GA-NAS on a weight-sharing supernet, outperforming other popular methods (DARTS, NAO, ENAS) using weight sharing, on the same NAS-Bench-101 search space. Also, the GA-NAS experiments on EfficientNet use a weight-sharing supernet.
> A few recent works on NAS have pointed out (e.g., Li and Talwalkar 19, Sciuto 19), an issue in NAS research is that there lacks fair comparison between different algorithms, given that they use different supernets or performance estimation schemes and different search spaces. Therefore, we provide extensive comparisons on public NAS benchmarks like NAS-Bench-101, 201 and 301 because they provide true accuracy and enable a fair comparison of the actual powers of the search algorithms. In real-world use cases, however, a user can choose freely how to search with GA-NAS either with or without weight sharing. And we demonstrated that using GA-NAS and weight sharing, we find a network that beats EfficientNet-B0.
>
>
> 2. Regarding your concerns on whether #Queries is a good metric for speed:
>
>  GA-NAS is a search algorithm which can be used with any performance estimation method including weight-sharing. Motivated by the fact that the performance estimation is expensive--- even with weight sharing, it is still costly to obtain the validation performance of a candidate architecture, a good “search algorithm” in NAS is the one which requires a low number of queries to its evaluator (whether it be a predictor or weight-sharing supernet) to reach to certain accuracy. This issue has sometimes been neglected in the literature as people tend to put search algorithm, search space and performance estimator together and only report the final accuracy.
> We dig into the power of the search algorithm and show that GA-NAS is one of the best search algorithm by requiring a fewer number of queries to locate the top architectures in these benchmarks.
> While the right #of queries to stop the search in reality can be set by checking whether the evaluator output stabilizes or simply run until a given budget is spent, the message we want to convey with the results on NAS Benchmark Datasets is that for the same or even less # of queries (budget), GA-NAS can find better architectures than the baselines on multiple search spaces, which in a real-world setting, would translate to better performance under the same or less cost in architecture evaluation. However, if weight-sharing is used, we did not report #Q in Table 3 or Table 6, following the same convention as other papers reporting weight-sharing NAS.

---

### Official Review · AnonReviewer4 · 2020-10-29

**Rating:** 6
**Confidence:** 3

**Review:**

Thanks for your informative response addressing my comments. After the revision, the description of the method is clearer (Sec 3.2), and the experimental results are clearer (Sec 4). I'll stay with my original accept-score.

===============

Summary:

The paper provides interesting results for neural architecture search. In particular, this paper proposes a search strategy for NAS problems, Generative Adversarial NAS (GA-NAS), using importance sampling, which can be applied to micro/macro, constrained/unconstrained search problems. GA-NAS beats the state-of-the-art search algorithms proposed for NAS on public benchmarks, including NAS-Bench-101, NAS-Bench-201, and NAS-Bench-301. Also, on the EfficientNet macro search space, GA-NAS finds a new architecture with higher ImageNet accuracy and a lower number of parameters than EfficientNet-B0.


Pros:

1. The proposed method achieves higher performance to compare to previous methods with better robustness, reproducibility, and efficiency.

2. The idea of NAS based on importance sampling for rare event simulation in the method seems interesting. The proposed method at the same time could be broadly applied to micro/macro, constrained/unconstrained search problems.

3. This paper provides comprehensive experiments, including various ablation studies, to show the effectiveness of the proposed framework.


Cons:

1. I suggest the authors conduct further ablation studies to enhance the understanding of the approach and readability of the paper:

(1) Comparison of computational resources (e.g. wall clock inference time) required for each query in Table 1 and Table 3. To be a fair comparison, it would be better to compare [number of queries * resource consumed per query].

(2) For the update algorithm of the generator, the proposed method uses JS-divergence minimization referring to [29]. Section 3.1 mentions that JS-divergence is more robust than KL-divergence, but I think a further analysis could strengthen the point.

(3) Adding FLOPs or inference speed to Table 5 and Table 6 would be helpful in explaining the performance of the new architecture found by GA-NAS.

2. Section 3, 4 need to be polished for better readability. For example, an explanation about the method and the concept of evaluation metric "rank" should be improved for the readers.


Some typos:

(1) In the section 3.2 “\tau=\{C_0,C_1,\ldotsC_{N-1}\}” -> “\tau=\{C_0,C_1,\ldots,C_{N-1}\}”
(2) In the equation (10) of appendix, “-(1-\rho) + P(X\leq \zeta^*)\geq 0.” -> “-(1-\rho) + P(X\leq \zeta^*)\geq 0,”

Some suggestions:
"\cdots"s are used in the expression such as "X_1,\cdots, X_N" and "S(X_1),\cdots, S(X_N)" in the proof of the theorem 6.2 in the appendix. I think "\ldots" is more syntactically typical here.

---

> ### Author Response · Authors · 2020-11-15
> **Computational Cost, inclusion of FLOP counts in Tables 5 and 6, and JS versus KL divergence comparisons.**
>
> Thanks for the constructive feedback on improving the writing of the paper.
>
> 1.	Computational cost:
>
> Regarding your questions on computational cost, we would like to note that the time needed for querying an architecture on NAS-Bench-101 or 201 does not vary between different search algorithms and is cheap. If we directly query the benchmark for performance then the computational cost is just the search time needed by the algorithm. The total search time of GA-NAS is about 8 GPU hours on NAS-Bench-101, including all the training of Discriminator and PPO, and querying the datasets. Other baselines do not report their wall-clock time on NAS-Bench-101. So it is difficult to compare this in a table.
>
> We believe # of queries is a fair metric for the resource consumption because in a real-world setting, if we do not use a weight-sharing supernet, then full-train evaluation is dominating. The number of queries made to the benchmark can directly represent the number of evaluations needed, thus approximating the total cost. Even if weight-sharing is used, it will still take some major time to train the supernet and evaluate each architecture using weights inherited from the supernet. In both cases, the total cost is mainly bottlenecked by evaluation cost.
>
> In the case of weight-sharing-based search in Table 3, GA-NAS takes about 1 GPU day for search and 1GPU day for supernet training on NAS-Bench-101, making a total cost of 2 GPU days. We’ll be adding this information to the paper.
>
> 2. Regarding your suggestion of showing the FLOPS of the found architectures in Table 5,6:
>
> We are looking into whether that is possible now. But we would like to clarify that for ResNet-constrained search, we used both the number of trainable weights and the training time as constraints, and for EfficientNet-constrained search we used the number of trainable weights as the constraint. Since our constrained search experiments did not incorporate FLOPS or inference latency as a constraint during search, it is unpredictable how they would compare to the original example model, i.e., the constraints we used may not correlate positively with FLOPS. The idea of Table 5, 6 is to show that GA-NAS is able to incorporate any (hard) constraints into search (including FLOPS/latency although not demonstrated here), and beat  the original model. So if one is interested in finding better architectures with lower FLOPS, he/she can easily include this as a constraint in the particular use of GA-NAS.
>
> 3. For JS Divergence vs KL Divergence: See the response to AnonReviewer3 titled “Explaining advantage of the symmetric JS Divergence”.

---

### Author Response · Authors · 2020-11-23
**Summary of Changes**

Dear reviewers,

We have uploaded a new version of the GA-NAS paper, as well as the source code for experiments. The major changes are summarized as follows:

First, we made several improvements to the experiment section (Sec.4), including the following:

1. New experimental results:
    1) Added new results for improving the ProxylessNAS-GPU model in Table 6 and A.3.9. GA-NAS was able to find a new network that beats the published ProxylessNAS-GPU model, with details on search space and the new model found provided in A.3.9.
    2) Added a Random Search baseline for constrained search on NAS-Bench-101 in Table 5.

2. Reorganized and rewrote Sec. 4.1 to better distinguish between weight-sharing and non-weight-sharing experiments, and showed that GA-NAS performs well in both settings. Re-organized Tables in Sec. 4.1 and added explanation to highlight the reproducibility and stability of GA-NAS by comparing mean + std and ranks achieved on public benchmarks.
Explained what is "rank", why rank is more important on these benchmarks can show reproducibility, and how the number of queries represents a comparison of search costs between algorithms.

4. Mentioned search time and computational cost of the GA-NAS experiments throughout Sec.4.

Second, we did a major revision to the method section (Sec.3) to improve clarity, including the following:

1. Re-wrote the entire Sec.3.2 to provide a clearer description on the proposed Generator and Discriminator models, and their training procedures. Moved some details to A.2. Pointed out that the Generator is not a complex model and its similarity to those used in ENAS and D-VAE.

2. At the end of Sec. 3.2, clarified the benefits and high efficiency of different components including the Discriminator in GA-NAS.

2. At the beginning of Sec. 3.2., improved explanation on how Algorithm 1 of Importance Sampling with JS Divergence minimization can lead to the proposed generative adversarial learning in GA-NAS (Algorithm 2).

3. Did a major revision to Appendix A.2 Model Details to improve clarity and illustration on model structures and their training procedure.

4. Added a paragraph in Appendix A.1 explaining the connections/differences between KL and JS Divergences, and why JS divergence minimization is used here.

Third, we have cleared an internal procedure and uploaded the source code of all experiments as part of the supplementary materials.

In addition, we fixed the typos, and all the citation, formatting and bibliography issues. Thank you again for pointing those out. We also revised the Abstract, Introduction, Conclusion and Related Work sections to further improve readability and clarity.

---

### Decision · Program_Chairs · 2021-01-07
**Final Decision**

**Decision:**

Reject

**Comment:**

This paper proposes a method for neural architecture search (NAS) based on adversarial methods. It uses a discriminator trained to distinguish between random vs. good architectures, letting the discriminator's scores serve as a reward signal for an autoregressive generator. I agree with AR1: this is a nice and clever idea. Reviewers generally agreed that the method was interesting, e.g. it's quite flexible in that it's able to incorporate constraints, and that the evaluation is rather extensive and shows that the method performs well across the board. Many minor criticisms were raised and addressed well by the authors in their responses and manuscript updates.

The major criticism shared by most reviewers was the high methodological complexity of the proposed approach, and the proportionally small gains shown over much simpler baselines. This criticism remained despite the authors' responses. The method is indeed complex: the same method without any adversarial component already performs well, and many important details of the model are relegated to Appendix A.2. (I would recommend, for example, moving Fig. 2 to the main text if at all possible. Also, the Appendix can/should be included in the main PDF for ICLR, rather than in supplementary material, as AR1 mentions.) It was not clear to reviewers that the adversarial component of the approach has a significant benefit. The authors respond by pointing to Table 7 showing that the discriminator reduces the number of queries and points out that in reality these queries correspond to expensive evaluations. If this is a major selling point of the method (it sounds like it could be), it should be highlighted and analyzed far more -- at least moved to the main text rather than an Appendix -- ideally with a real-world evaluation showing a practical large improvement in overall wall-clock time, rather than a benchmark where these evaluations are free. Perhaps the exclusive reliance on these benchmarks, though undoubtedly useful for quick experimentation, in the end holds back the paper and prevents the method's benefits from becoming apparent to the readers.

As a minor point (also raised by AR1), the paper is formatted incorrectly for ICLR: the font color is off, and more importantly the PDF is unsearchable (text cannot be selected, ctrl-F does not work), which makes it very difficult to quickly reference and review. Please try not to stray from the conference-provided style file for future submissions.

I appreciate the cleverness of the method, the extent of the evaluation, and the thorough responses to the reviews. However, unfortunately with the current presentation, it is too difficult to discern the benefit of the proposed approach from the manuscript. The approach is nonetheless intuitively appealing and seems quite promising, and I hope the authors will take the reviewers' good feedback into account and resubmit the paper in the future.